# The Use of Pluripotent Stem Cell-Derived Organoids to Study Extracellular Matrix Development during Neural Degeneration

**DOI:** 10.3390/cells8030242

**Published:** 2019-03-14

**Authors:** Yuanwei Yan, Julie Bejoy, Mark Marzano, Yan Li

**Affiliations:** Department of Chemical and Biomedical Engineering, FAMU-FSU College of Engineering, Florida State University, Tallahassee, FL 32310, USA; yyan73@wisc.edu (Y.Y.); julie1.bejoy@famu.edu (J.B.); mcm12g@my.fsu.edu (M.M.)

**Keywords:** extracellular matrix, pluripotent stem cells, organoids, neural degeneration, three-dimensional

## Abstract

The mechanism that causes the Alzheimer’s disease (AD) pathologies, including amyloid plaque, neurofibrillary tangles, and neuron death, is not well understood due to the lack of robust study models for human brain. Three-dimensional organoid systems based on human pluripotent stem cells (hPSCs) have shown a promising potential to model neurodegenerative diseases, including AD. These systems, in combination with engineering tools, allow in vitro generation of brain-like tissues that recapitulate complex cell-cell and cell-extracellular matrix (ECM) interactions. Brain ECMs play important roles in neural differentiation, proliferation, neuronal network, and AD progression. In this contribution related to brain ECMs, recent advances in modeling AD pathology and progression based on hPSC-derived neural cells, tissues, and brain organoids were reviewed and summarized. In addition, the roles of ECMs in neural differentiation of hPSCs and the influences of heparan sulfate proteoglycans, chondroitin sulfate proteoglycans, and hyaluronic acid on the progression of neurodegeneration were discussed. The advantages that use stem cell-based organoids to study neural degeneration and to investigate the effects of ECM development on the disease progression were highlighted. The contents of this article are significant for understanding cell-matrix interactions in stem cell microenvironment for treating neural degeneration.

## 1. Introduction of Alzheimer’s Disease Pathology

### 1.1. Alzheimer’s Disease Pathology and Progression

Alzheimer’s disease (AD) is a very common, incurable age-associated and economically costly disease characterized by progressive neurodegeneration, which causes deterioration and damage of neurons within the cerebral cortex, loss of memory, and cognitive decline [1]. It is the most common type of dementia and over 30 million people are suffering from this disease all over the world [2]. About 500,000 people die of AD every year and this number increases year by year. Currently, no effective treatments or available drugs can cure AD and the total costs in 2010 were around $172 billion in the United States [2], which makes it a heavy economic burden not only on the dementia patients but also on our global society. Most of clinical cases in AD develop late-onset symptoms (after the age of 65), also called sporadic AD (SAD). And about 2–5% of disease burden is an early-onset type, called familial AD (FAD). FAD is related to genetic mutations, such as mutations in amyloid precursor protein (APP), presenilin-1 (PS1), and presenilin-2 (PS2) [3]. The gene of PS1 encodes the catalytic subunit of γ-secretase that mediates APP cleavage for the generation of amyloid β42 (Aβ42) peptides [4].

AD is identified by three cardinal features in human brain: Aβ plaques, neurofibrillary tangles (NFTs), and neuron loss. Aβ plaques are usually formed because of the increased production or deficient clearance of neuronal Aβ caused by unknown reasons. APP is proteolyzed to the soluble and nontoxic Aβ monomers by β- and γ-secretases. The monomeric Aβ peptides then aggregate to form toxic Aβ oligomers and further generate the insoluble fibrils, which ultimately form the plaques (Figure 1) [5]. Some extracellular matrix (ECM) components, e.g., heparan sulfate proteoglycans (HSPG), are considered to promote extracellular formation of amyloid plaques [6]. In addition, a healthy body has the mechanism to remove Aβ peptides and balance their generation and clearance. For instance, matrix metalloproteinase 3 (MMP3), a collagen IV proteinase, could degrade Aβ components [7,8]. Also, a defective clearance of Aβ and an unbalance between Aβ generation and clearance have been demonstrated in SAD [9]. Thus, the peculiar generation and flawed removal of Aβ peptides cause Aβ accumulation. The more accumulations of Aβ peptides occur, the more senile plaques are formed in the human brain.

The NFTs in AD are composed of paired helical filaments consisting of hyperphosphorylated tau (p-tau), a protein associated with the microtubules [10]. The normal form of tau is soluble, whereas it depicts strong ability to assemble and become insoluble under physiological conditions. The over phosphorylation of tau contributes to its disassembly from microtubules and its aggregation to form insoluble fibers, that is, the NFTs. Protein kinases, such as mitogen-activated protein kinase (MAPK), glycogen synthase kinase-3 (GSK-3), and Jun N-terminal kinase/stress-activated protein kinases (JNK/SAPK), could upregulate phosphorylation of tau. Many studies support the notion that Aβ may interplay with tau-derived NFT formation, while critical questions about Aβ-induced tau pathology in AD are still unanswered. Overall, the perturbation of tau kinases and tau phosphatases leads to the abnormal hyperphosphorylation of tau, which results in NFT generation and toxicity in neurons.

The Aβ plaque and NFT formation show toxicity to neuronal cells. However, the causes of synaptic dysfunctions and selective loss of vulnerable neurons, particularly the subtypes, such as pyramidal, cholinergic, noradrenergic, and serotonergic neurons, remain elusive [11]. Studies show that Aβ plaques could trigger an inflammatory process by increasing the secretion of pro-inflammatory factors, e.g., IL-6 and TNF-α, which impair microglia and the surrounding neural cells, and finally result in neurodegeneration and neuron loss [12,13]. Based on the Aβ plaque deposition and cortical NFT formation, AD initiates when the connection between entorhinal cortex and hippocampus begins to disappear at the transentorhinal stage [5,14]. Other neuronal cells, such as GABAergic interneurons, show their function loss during the advanced stage of AD, i.e., the isocortical stage.

### 1.2. Current Challenges and the Demand for a Good AD Model

Although the pathophysiological changes of AD are characterized, it is unclear what factors induce the disease and how selective neuronal loss in AD is caused. Many hypotheses have been proposed for the causes of pathological characteristics [5,9]. These hypotheses include: (1) amyloid hypothesis, which posits that the gene mutations of APP and PS1/2 induce the overexpression or deficient removal of Aβ toxic species, and thus lead to the other two cardinal changes associated with AD (Figure 1); (2) tau hypothesis, which believes that the hyperphosphorylated tau protein disassembles from microtubules and subsequently induces neuronal death (Figure 1); (3) unknown triggering hypothesis, which postulates that some uncertain factors lead to the neurodegeneration, both directly and indirectly, through the formation of plaques and tangles. The last case seems to incorporate both the amyloid and tau hypotheses to interpret the etiology of AD. Although both amyloid and tau hypotheses could explain some of the pathological hallmarks and clinical observations about AD, there is still a controversy over which one occurs first. Thus, it is necessary to establish the disease models of AD, given that it is difficult to obtain patient brain samples.

To date, the modeling of AD for mechanistic and pathological studies mainly relies on non-human animal models and non-neuronal human cells. Non-neuronal cells usually lack the specific cellular structures of neurons and fail to recapitulate the signaling pathway and other physiology of neurons. Therefore, they cannot capture the three pathological changes in AD [15]. The transgenic animal models, usually the mice, have been developed for discovering genetic mutations in FAD [16,17,18]. To some extent, these models could interpret the pathogenicity of Aβ accumulation, plaque formation, and tauopathy. For instance, one study has developed a transgenic AD mouse model with a mutation in APP [17]. The model showed senile plaque formation and disconnection of synapses, but the presence of NFTs and neuronal loss was never found. Although animal models displayed the benefits for understanding AD neuropathology, they fail to reflect human brain anatomy, genetics, and the AD-associated neuron loss. More importantly, pharmacological testing and candidate drug target screening for AD using these animal models have shown no successful development for AD therapeutics till now [19]. Therefore, a more robust and suitable model that could capture all the three cardinal characteristics of AD is demanded. Recently, human adult neural stem cell-derived models through 3D culture in vitro, although limited by the cell source, were found to increase Aβ plaque depositions and tau phosphorylation [20,21]. In another study, Choi et al. reported a 3D thin-layer culture system by embedding human neural stem cells into Matrigel to model AD [22]. The overexpression of FAD mutations in APP and PS1 in embedded cells could induce both extracellular Aβ plaque and phosphorylated tau aggregation in the soma and neurites. These studies accurately paved the way for using human induced pluripotent stem cells (hiPSCs), a more reproducible and scalable cell type, to model AD in vitro.

## 2. Human Pluripotent Stem Cells for Modeling AD

Human PSCs (hPSCs) include human embryonic stem cells (ESCs) derived from blastocysts and hiPSCs reprogrammed from somatic cells. HPSCs show the properties of unlimited self-renewal and potent differentiation capacity. Theoretically, hPSCs could give rise to the three-germ layers, including ectoderm, endoderm, and mesoderm, and generate every cell type in the body (e.g., neurons, heart, pancreatic, and liver cells). HiPSCs were pioneered by Shinya Yamanaka’s lab in 2006 that the introduction of four reprogramming factors including Oct4, Sox2, c-Myc, and Klf4 (also dubbed Yamanaka factors) could convert adult somatic cells into PSCs [23]. Using hiPSCs to model AD shows great advantages compared to other cell lines or model systems. First, patient-specific cell lines can be easily established just through collecting biopsy from patients (e.g., drops of blood or a piece of skin) and reprogramming the tissue to hiPSCs (Figure 2). Therefore, the ethical controversy using human embryos for human ESCs can be avoided. For AD, the patient-specific hiPSCs loaded with gene mutations can easily recapitulate the mechanism of disease progression, especially for the early-onset type. Thus, the transgenic technology for specific genetic manipulation used in animal models becomes unnecessary. HiPSCs are also clinically advantageous since the use of autologous tissue ideally surpasses the patient’s immune rejection. Moreover, hiPSCs with unlimited self-renewal capacity could help to provide large numbers of patient-specific neuronal cells for in vitro research and clinical applications.

### 2.1. AD Models Using hPSCs

Generally, three methods are utilized to establish AD pathological phenotype using hPSCs (summarized in Table 1). The first method is chemical induction, in which extrinsic chemicals, such as Aβ42 oligomers and aftin5 (an Aβ42 inducer), are used to treat healthy hPSC-derived neural cells and induce AD-related phenotypes [24,25,26]. The addition of extrinsic chemicals to the normal hPSC-derived neurons could recapitulate some AD events, such as cytotoxicity of neuronal cells. However, the progression and some important AD pathologies, such as extracellular Aβ plaque formation, are difficult to recapitulate. The second method is to use genetic tools, such as lentivirus vectors or CRISPR-Cas9 gene-editing system, to overexpress AD-related genes, such as APP, PS1/2, and apolipoprotein E3/E4 (APOE3/E4), in hPSC-derived cells [27,28,29]. Using human ESC-derived neurons with recombinant human APOE2, APOE3, or APOE4, Huang et al. showed that all three APOE isoforms could upregulate APP and Aβ production but with different efficacy (APOE4 > APOE3 > APOE2) [29]. Park et al. co-cultured neurons and astrocytes derived from hPSCs overexpressing FAD-APP mutations with immortalized SV40 microglia in a microfluidics-based system to study neuro-inflammatory activity in AD [30]. For this approach, overexpression of mutant proteins is needed to induce AD phenotype.

The third method is to use AD patient-specific iPSCs carrying AD phenotype and mutations [42,59]. HiPSCs derived from familial and sporadic AD patients have been differentiated into various types of neuronal cells for studying specific AD pathologies (Table 1). The patients generally had genetic mutations in APP, PS1, or PS2, in the case of FAD, and mutations in APOE3/E4 for SAD. Most of the hPSC-derived AD models used either 2D or embryoid body/neurosphere differentiation protocols to generate forebrain neurons, such as cortical glutamatergic neurons, GABAergic interneurons, or cholinergic neurons. The neuronal cells were characterized through gene and/or neuronal marker expression and tested for action potentials and calcium-handling ability for functional assessments. The AD-related pathology including elevated Aβ42 production and hyperphosphorylated tau was indicated in these models [71,72].

There are two main classes of mutations for familial AD: (1) mutations in APP; and (2) the mutations in PS1 or PS2 [73]. Aβ is a cleavage product of APP, so the dysfunction of APP processing can cause AD (Figure 1). PS1/PS2 mediate the regulated proteolytic events of several proteins including γ-secretase [39], which plays an important role in Aβ generation. Mutations in PS1/PS2 also lead to Aβ plaque accumulation. Human iPSC lines were derived from familial (APP mutation) and sporadic AD patients and differentiated into cortical neurons (expressing TBR1 and SATB2) [37]. In FAD-derived cells, extracellular Aβ42 increased along with the decrease in intracellular Aβ2 compared to control neural cells. For SAD-derived cells, extracellular Aβ42 did not change while intracellular Aβ42 decreased compared to control of neural cells. Astrocytes also accumulated intracellular Aβ, which increased endoplasmic reticulum (ER) stress and reactive oxygen species (ROS). To improve ER stress and inhibit ROS generation, drugs including BSI, DHA, NSC23766, and dibenzoylmethane were evaluated. DHA decreased BiP (an HSP70 molecular chaperone located in the lumen of the ER) protein level, cleaved caspase 4 and peroxiredoxin-4, and decreased ROS generation. Differential response to DHA was observed for different patient-specific hiPSCs, indicating that DHA may be used for a subset of AD patients.

To evaluate if hiPSCs can recapitulate AD phenotype in the patients and understand the relationship between APP processing and tau phosphorylation, hiPSCs were derived from two familial (mutations in APP) and two sporadic AD patients [36]. Neurons from AD patients (two familial and one sporadic) had elevated levels of Aβ40, p-tau, and active glycogen synthase kinase-3β (aGSK-3β). β-secretase inhibitors, but not γ-secretase inhibitors, reduced p-tau and aGSK-3β levels. In AD, synaptic loss is one pathological observation, and the decreased synapsin I is usually observed in the affected human brain. In this study, no synaptic loss was observed and an extended culture period may be required to observe synaptic loss.

Neural progenitor cells (NPCs) derived from hiPSCs from PS1 FAD patients were also characterized [39]. The differentiation from hiPSCs used monolayer-based protocol with dual SMAD inhibition for 14 days. The derived NPCs displayed normal electrophysiology. However, an increased ratio of Aβ42/Aβ40 was observed in the conditioned medium compared to the control cell lines. Genetic profiling identified the genes that were differentially regulated due to PS1 mutations in order to find the molecules that might have a developmental role in FAD pathology. Three targets were found, including NLRP2, ASB2, and NDP. In particular, NDP expression decreased in the hippocampus of the late-onset AD brains. Moreover, synaptic dysfunction was observed in hiPSC-derived cortical neurons with autosomal dominant AD mutations or trisomy of chromosome 21 [58]. The released Aβ peptides from the FAD neurons with APP or PS1 mutations blocked hippocampal long-term potentiation (LTP), while neurons with trisomy 21 inhibited LTP through extracellular tau [58]. In another study, APOE4-expressing neurons showed elevated levels of tau phosphorylation, which was not related to the increased Aβ production, and the cells displayed GABAergic neuron degeneration [66].

### 2.2. Neural Tissue Patterning of hPSCs

The advancements using hPSCs to model AD started with the generation of neuronal cells or tissues that are directly affected by the disease through neural patterning (Figure 2). The key to success of this utilization is to efficiently generate specific neuronal subtypes or brain-like tissues from hPSCs. Neural tissue patterning is a complex process governed by intrinsic and extrinsic factors including morphogens and cell-ECM interactions. During the development of mammalian brain, early neural progenitors of the neural tube are derived through the inhibition of bone morphogenetic protein (BMP) and activin signaling. The cells are then specified into neuronal subtypes of each brain region along the rostral-caudal axis by tuning Wnt and retinoic acid (RA) signaling pathways, and along the dorsal-ventral axis mainly by the gradient of sonic hedgehog (SHH) proteins. Based on this knowledge, protocols utilizing the morphogens to mimic brain development have been established to produce brain region-specific neuronal subtypes derived from hPSCs (Figure 3). For example, cortical glutamatergic neurons and GABAergic neurons from hPSCs were acquired by the modulation of SHH and fibroblast growth factor (FGF)-2 signals [25,74], while hindbrain/spinal motor neurons were efficiently derived through the activation of SHH, RA, and Wnt signaling pathways [75]. HPSC-derived spherical brain-like tissues, such as cerebral organoids with definable forebrain, midbrain, and hindbrain/spinal cord layers [76], and cortical spheroids with laminated regions [77] were generated by the default neural development without extrinsic factors. Interestingly, recent studies have obtained the whole spectrum of brain region-specific neural progenitors and neuronal subtypes from hiPSCs [78,79], which indicates the remarkable value of hiPSC-based models for the study and treatment of patient-specific neurological diseases. For AD, the Aβ deposition, tau tangles, and neuronal loss ultimately occur in the cerebral cortex, leading to the cortical symptoms related to language, attention, and visuospatial orientation. Therefore, studies modeling AD with hiPSCs often begin with the generation of forebrain cortical neurons. Differentiation to forebrain neuronal fates is based on the “default” induction strategies without exogenous patterning factors after dual-SMAD inhibition in a monolayer culture or embryoid body (EB)-based culture [77,80].

### 2.3. A Novel Neural Patterning Method: Organoid Technology

The organoid systems derived from EB-based cultures emerge as a mixed cell population in a 3D platform. Relying on the intrinsic ability of self-organization of hiPSCs, and sometimes with the help of suitable exogenous factors (e.g., Matrigel), organoids recapitulate a large number of biological events in vivo [82,83]. Organoids are 3D spatial organization including heterogeneous tissue-specific cells, cell-cell interactions, and cell-ECM interactions, and exhibit certain physiological functions similar to tissues or organs in human body [84]. During organoid formation, ECM is one of the most important patterning factors that regulate and assist the self-organization and differentiation of stem cells within the organoids (Figure 4) [85]. ECMs, either secreted by stem cells or derived from artificial scaffolds, provide physical supports for cell attachment and organization and additional supplementation of signaling cues for cell growth and differentiation [86,87]. This organoid technology bridges a gap in the existing in vitro 3D culture systems by providing a robust approach for tissue patterning of stem cells and is more representative of in vivo situations [88,89,90].

The first generation of neural rosettes from human ESCs was derived in 2001 by forming spontaneously differentiated EBs and then plating EBs on coated dishes for neuroepithelial cluster formation [91]. Combining the EB-based rosette patterning and serum-free culture, the so-called SFEBq approach, i.e., serum-free floating culture of EB-like aggregates with quick re-aggregation, was developed and could generate remarkably large rosettes with lumens and apicobasal structures from hPSCs in 2008 [92]. In 2011, the self-organizing optic cups with retinal specifications from hPSCs were derived using the entirely 3D EB-based neural culture [93]. This study indicates that neural tissues with histologically accurate architecture could be patterned only in 3D floating culture of hPSCs. All these studies paved the road for the advent of a new neural patterning technology, that is, organoid. In 2013, Lancaster et al. derived cerebral organoids with a broad brain regionalization to model microcephaly using EB-based culture with Matrigel embedding and agitation [76]. The organoids could reach up to 4 mm in diameter with fluid-filled cavities that resembled ventricles rather than the small neural-tube-like lumens found in rosettes. After that, the organoid-based neural patterning methods of hPSCs have been utilized for mimicking human brain development, modeling neurological disease in vitro, and testing potential drug candidates [81].

Using the SFEBq method, multilayer structures were derived from hiPSCs with lower and upper cortical layer fates in dorsal forebrain [94]. The 3D aggregates can generate inhibitory neurons and display neuronal connectivity (expression of synapsin I and PSD95). Genomics study showed the correlation to the brain development at postconceptional weeks 4–10. Another 3D culture was used to generate laminated cerebral cortex-like structure from hPSCs [77]. The cells were induced for forebrain fate using dual SMAD inhibition for six days, followed by FGF-2/epidermal growth factor treatment for 19 days, and the maturation with brain-derived neurotrophic factor (BDNF) and NT3. At day 18, 85% PAX6+ cells and >80% of FOXG1+ cells were observed. The organoid size increased from 300 µm at two weeks to 4 mm at 2.5 months. The formed human cortical spheroids corresponded to the late mid-fetal periods. The spheroids had a distinct ventricular zone and sub-ventricular zone. The cells also formed superficial cortical layers (expressing BRN2 and SATB2 for layer II-IV, emerged at day 76) and deep cortical layers (expressing CTIP2-layer V and TBR1-layer VI, showed up at day 52). The presence of GFAP+ cells showed the astrogenesis (3–8% cells). Functional characterizations include: (1) calcium imaging for spontaneous calcium spikes; (2) Na^+^/K^+^ currents; and (3) the firing of action potentials. Slices of spheroids showed spontaneous synaptic activity, which can be reduced by a glutamate receptor blocker. This study demonstrated the patterning and specification of different neuronal and glial cell types which can be used for large-scale drug screening.

The organoid technology is a promising method to mimic human brain development and could generate brain-like tissues for modeling neurological diseases, including autosomal recessive primary microcephaly [95], Zika virus infection [96], autism spectrum disorder [97], Timothy syndrome [98], brain tumors [99,100], and others (see the review by Amin et al., 2018 [101]). Lots of studies have shown the feasibility of the organoid methods to model AD pathogenesis [24,52,60,61]. Our previous studies found that dynamic culture of cerebral organoids promoted cortical layer structure compared to static culture, and the 3D brain organoids from hiPSCs with PS1 M146V mutation could recapitulate some AD-related phenotype, such as Aβ secretion and neuron death [59,102]. By using hiPSCs with APOE4/E3 mutations, Lin et al. showed that APOE4 forebrain organoids displayed the increased Aβ aggregation and hyperphosphorylated tau [61]. Taken together, 3D hPSC-derived organoids provide structured neural tissues and specific microenvironments that can be used to model AD-associated pathology.

### 2.4. Effects of ECMs on Neural Patterning of hPSCs

The central nervous system (CNS) is characterized by a functional network of neurons, glia, and their secreted ECMs. ECMs constitute the essential physical structures for neural cells and provide diverse biochemical signals for neurogenesis, neural cell migration and differentiation, and synaptic plasticity. ECMs also play an important role in neural tissue patterning of hPSCs. Studies performed with decellularized ECMs from different PSC derivatives indicated that ECMs guided PSC differentiation into the cell types residing in the tissue from which the ECMs were derived [103,104]. The ECM-stem cell interactions are mainly mediated by integrins, a large family of heterodimeric transmembrane receptors connecting with an intracellular cytoskeleton. Thus, cell migration, organization, and differentiation could be regulated through ECM-integrin-cytoskeleton connections. For instance, the binding between α6β1 integrin and laminin is the key player for neural stem cell adhesion to the vascular structures [105]. ECMs may also control stem cell fate decisions through the modulation of intracellular signaling. For example, our previous study demonstrated that undifferentiated PSC-derived ECMs could upregulate Wnt/β-catenin signaling, while NPC-derived ECMs promoted neural patterning of PSCs through the inhibition of Wnt signaling pathway [106].

The biophysical properties, such as elastic modulus, geometry, and Poisson’s ratio of matrix, also impact growth, proliferation, and neural differentiation of PSCs [107,108]. Stem cells cultured on hydrogels with varied stiffness indicated that substrate elastic modulus can alter critical cellular events, such as ECM assembly, cell motility, and cell spreading [109,110,111]. For neural patterning, a large number of studies showed the important effects of matrix stiffness or elastic modulus (Table 2) [111,112,113,114,115,116,117,118]. Saha et al. first showed that soft substrates supported neuron differentiation from NPCs, while a higher elastic modulus (*E* ~ 1 to 10 kPa) promoted glial cell generation [111]. Leipzig et al. further demonstrated that substrates with Young’s modulus (*E*) below 1 kPa favored neuronal differentiation, those in the range of 1 and 3.5 kPa supported astrocytes, and above 7 kPa could enhance oligodendrocyte derivation [112]. By culturing hPSCs on Matrigel-coated polyacrylamide substrates, a very soft matrix (*E* ~ 0.1 kPa) was found to support early neural differentiation of hPSCs [119]. Normally, cells sense elasticity during the attachment on the substrate through focal adhesions and formation of stress fibers. Their responses to the matrix properties rely on myosin-directed contraction and cell-ECM adhesions, which involve integrins, cadherins, and other adhesion molecules [120]. The Poisson’s ratio is another important biophysical cue that influences stem cell behaviors, as the nuclei of ESCs exhibit a negative Poisson’s ratio in the pluripotent-state [121]. Our previous work found that Poisson’s ratio of matrix could confound the effects of elastic modulus on PSC neural differentiation [108]. In conclusion, ECMs serve as a reservoir of biochemical and biophysical factors that impact stem cell growth, organization, and differentiation.

## 3. Proteoglycans in the ECMs of AD Brain

The brain ECMs express low amounts of fibrous proteins, such as collagen and laminin, but high amounts of glycosaminoglycans (GAG) compared to other tissue types. The ECMs in the brain include glycoproteins and proteoglycans originated from neurons or glia. Proteoglycans (PGs) are a group of glycoproteins that carry covalently bound sulfated GAG chains which play important roles in cell differentiation, tissue morphogenesis, and phenotypic stabilization via cell-matrix adhesiveness and/or binding to growth regulators. GAGs are composed of repeating disaccharide units attached to a core protein through serine residue and carbohydrate linkage regions. Depending on the disaccharide structures, PGs can be divided into chondroitin/dermatan sulfate, heparin/heparan sulfate (HS), and keratan sulfate side chains. In the CNS, the majority of proteoglycans have either heparin sulfate or chondroitin sulfate side chains. PGs are expressed at different stages of human brain development and are produced non-homogenously by different neural cell types [144].

### 3.1. Chondroitin Sulfate Proteoglycans (CSPGs) in Brain Development

CSPG family consists of four members: (1) lecticans, a family including aggrecan, versican, neurocan, and brevican; (2) phosphacan; (3) small leucine-rich proteoglycans, e.g., decorin and biglycan; (4) other CSPGs including neuroglycan-C and NG2 [145]. Distribution and expression of these CSPGs vary during human brain development. Aggrecan and versican are distributed in ECMs of various non-neuronal tissues as well as neural tissues, whereas neurocan and brevican appear to show a neural tissue-specific distribution. Versicans are found to express in the post-natal development of brains. Brevican is expressed predominantly by astrocytes, not neurons. NG2, a large transmembrane CSPG, is expressed by oligodendrocyte progenitor cells in CNS. Neurocan, a secretory CSPG, is reported to be synthesized mainly by neuronal cells.

The CSPGs are either secreted as ECMs or inserted to the plasma membrane of the cells. The attachment of CSPGs to the cell membrane is either through a collagenous ligand or by core proteins and cell surface-associated CSPGs such as aggrecan. The concentrations of aggrecan and brevican increase during brain development. The concentration of neurocan also increases with development to reach a peak in the early postnatal period and declines thereafter. The full-length neurocan is the major variant in pre- and early postnatal brains, while it is hardly detectable in the mature brain. Phosphacan and receptor-type protein tyrosine phosphatase are produced by both glial cells and neuronal cells. Variations in sulfate content and localization on the same GAG chains or the same core protein are tissue-specific or cell-specific. The expression of specific GAG biosynthetic and modifying enzymes by individual neural cell types dictates the type of carbohydrate modification of the core proteins expressed in each cell type [146].

### 3.2. Heparin/HSPG in Brain Development

Heparan sulfate proteoglycans are heavily glycosylated proteins, in which some HS and GAG chains are covalently attached to a core protein, which can be either surface proteins (e.g., syndecans/glypicans) or secreted proteins (e.g., agrin/collagen type XVIII/perlecan) [147]. N-Syndecan (syndecan-3), a transmembrane HSPG, has the capacity to bind to heparin-binding growth factors with a neurite-promoting activity, such as FGF-2, pleiotrophin/HBGAM, and midkine. This matrix-growth factor interaction is found to be the basis for the axonal development of neural cells. Also, N-syndecan works as a receptor molecule for pleiotrophin, which then enhances the phosphorylation of intracellular cytoskeleton-regulating molecules, such as cortactin and fyn-kinase. It was found that the association of cortactin and fyn-kinase with N-syndecan was increased after induction of long-lasting synaptic sensitivity called long-term potentiation.

N-syndecan regulates the neuronal activity-dependent connectivity through the fyn signaling pathway. N-syndecan, together with syndecan-2, also appears to be involved in the maturation of synapses. Syndecan-2 interacts with calcium/calmodulin-dependent serine protein kinase, a PDZ family protein that induces the formation of mature dendritic spines, and supports the development of postsynaptic specialization [148]. Heparin is one of the glycosaminoglycans that bind to different proteins and thereby promote various neural functions, such as preventing blood clotting, neuronal communication, and brain development.

### 3.3. Impacts of CSPG on AD Pathology

The motor and sensory areas of the cortical brain, which show cytoskeletal changes during AD development, were found to express prominent levels of CSPGs. These results suggested the importance of CSPGs on AD development, e.g., synaptic loss. Numerous molecular events occur in the post-synaptic density of an excitatory synapse in response to a train of pre-synaptic action potentials. One of these factors is the presence of peri-synaptic ECMs, a lattice of chondroitin sulfate-bearing proteoglycans, termed lecticans (e.g., aggrecan, brevican, neurocan, and versican), bound to hyaluronic acid near their N-terminus and tenascin-R near their C-terminus. Alterations in brain ECMs occur early in AD progression. The synaptic loss is observed prior to neuronal cell death, and the loss of synapses in the outer molecular layer of the hippocampal dentate gyrus is more highly correlated with cognitive impairment than other classical AD pathology including NFTs and senile plaques. ECMs can respond to network activity by incorporating secreted molecules, by shedding extracellular domains of transmembrane molecules, or by freeing products of activity-dependent proteolytic cleavage as signaling messengers.

Lecticans, also known as hyalectans, have been studied for their influence on the plasticity of synapses because of the elevated expression in AD patients. Lecticans are found to contribute to diminished synaptic plasticity. Involvement of CSPGs in hippocampal synaptic plasticity was confirmed using the enzyme that specifically digests chondroitin sulfates, chondroitinase ABC (ChABC) [149]. Injection of ChABC in young AD mice elevated lectican levels and resulted in a reversal of contextual fear memory deficits and restoration to normal long-term potentiation. ChABC treatment results in the removal of CS chains from lectican core proteins (and other CS bearing PGs). These studies demonstrated that removal of CS chains stimulated neural plasticity. Increasing evidence has demonstrated that CS-bearing ECM molecules increase with age and are associated with AD. CS-bearing PGs bind to Aβ, but it remains unknown whether Aβ binds to brain-derived lecticans. Injection of ChABC increased synaptic density surrounding plaques and reduced Aβ burden in the s/m of the hippocampus [150].

### 3.4. Impacts of Heparin/HSPGs on AD Pathology

HSPGs are found to grow concomitantly with amyloid filament formation. HSPGs have a negative charge and thereby concentrate on the ligands and protect them from proteolysis while making insoluble complexes [151]. Heparin and HSPGs regulate the activity of a number of proteases involved in AD. HSPGs promote Aβ aggregation, stabilize Aβ fibrils, and inhibit Aβ degradation. Perlecan and agrin are found to be the important HSPGs that are involved in the pathogenesis and localized with Aβ deposits in AD brain. Perlecan accelerates Aβ fibril formation and maintains the Aβ fibril stability by using Perlecan’s HS-GAG chains [152]. Agrin also binds to Aβ and accelerates fibril formation and protects fibrillar Aβ from proteolysis (Figure 5) [153]. HSPGs bind Aβ with high affinity and promote intracellular Aβ uptake in multiple cell types and thereby increase the cytotoxicity [154].

HSPGs are also the target of Aβ-induced oxidative stress production. Studies propose that minimal chain length is a prerequisite for efficient fibril polymerization and deposition. Fragmentation of HS using heparinase III was found to inhibit the polymerization and deposition of amyloid peptides [155]. Heparinase is a glucuronidase that specifically cleaves heparan sulfate, thereby preventing Aβ-HSPG interactions. The formed HS oligosaccharides then bind amyloid monomers to prevent them from polymerizing and assembling into larger aggregates. Treatment of cells with heparinase III also reduced the HSPG-based Aβ-induced ROS production.

HSPGs modulate the Aβ-associated neuroinflammation and Aβ clearance from the human brain. Several in vitro studies indicated the interaction of Aβ with GAGs including HS and heparin, a HS analogue with a higher sulfation degree [156]. All those studies describe that the N-terminus of Aβ is a HS-binding motif and this sequence supports the interaction between Aβ and HSPGs [157]. Studies also suggest that the Aβ peptides compete with FGF-2 for HS binding site. The interaction depends on several factors, such as the sulfation pattern, chain length of the GAGs, and the pH. The higher sulfation in heparin boosted the affinity and the degree of Aβ-HS complex formation was higher compared to Aβ-HSPG complex. Lower binding of heparin fragments shorter than 6-sugar units to Aβ also suggested the requirement of GAG chain length for the interaction [158]. It is also proposed that the Aβ-HS interaction is mutually protective because HS is protected from heparanase degradation and Aβ is protected from protease degradation.

Cellular degradation of HSPGs using heparanase (an endo-β-glucuronidase that specifically cleaves HS side chains of HSPGs) decreased the cytotoxicity of Aβ peptides on the cells [154]. Aβ clearance from the cells is performed in different ways, including degradation by proteolytic enzymes or receptor-mediated Aβ transport across the blood-brain barrier, in which the main receptor is low-density lipoprotein receptor-related protein-1 (LRP-1) [159,160]. The complex interactions of Aβ precursor protein ApoE, LRP-1, and HSPG facilitate Aβ internalization. Recent studies found that HSPGs and LRP-1 mediate Aβ internalization in a seemingly cooperative manner, in which HSPGs regulate Aβ binding to cell surface through LRP-1 [159]. These findings suggest that cell surface HS mediates Aβ internalization and toxicity. Tau binding and internalization also depend on HSPGs, which are critical mediators of tau fibril internalization. Tau binding to HSPGs is required for transcellular propagation [161]. Further understanding of the interactions between ECM molecules and Aβ peptides and their effects on neuronal differentiation should help to develop new treatment methods for AD.

### 3.5. Heparin-Based Therapy for Neural Degeneration

Aβ-related activation and the followed neuroinflammation are dependent on the region 1–11 of the peptide [162]. Related studies of the molecules that have the ability to pharmacologically target this region indicate the possible therapeutic effects of heparin [163]. In vitro treatment of neurons with heparin reduced Aβ-associated cytotoxicity, revealing the capability of heparin on AD treatment. Heparin, which has been widely used as an anticoagulant in clinical use, has the capability to bind to the region 12–17 of Aβ (HHQK) [157], near to the contact and complement activation residues. The heparin-binding regions of these proteins are characterized by clusters of arginines and lysines. These clusters form centers of highly positive charge density that electrostatically interact with the acidic groups of heparin.

The binding of heparin to the regions of Aβ occurs as electrostatic interactions between cationic sites of Aβ and anionic sulfate residues of heparin. This causes steric changes in the Aβ and influences the biological function, which in turn reduces the inflammation of neural cells by inhibiting Aβ complement and contact systems and thereby reduces the neurodegeneration. Complement and contact systems are related inflammatory cascades and are dependent on the reciprocal activation. These systems share a major inhibitor factor (C1 INH) with each other (Figure 6), which is downregulated in AD brain. Inhibition of complement activation by heparin might also be due to the potentiation of C1-1NH. C1-1NH attaches to the enzymes and works as an inhibitor of spontaneous activation of the complementary system. The addition of low molecular weight heparin reduced plaques and Aβ accumulation in a mouse AD model [164]. Association between heparin and the Aβ peptides was found to be pH-dependent. Heparin-binding to the peptides reduced the concentration of the peptides and the expression of kinnogen.

Biologically active beta-site amyloid precursor protein cleaving enzyme 1 (BACE1) interacts directly with HS/heparin and has a high relative affinity for these polysaccharides. It has been reported that heparin inhibits BACE1 activity in vitro and thereby regulates the production of Aβ peptides [163,165]. It is also suggested that Aβ is a metalloprotein that expresses both high and low metal affinity sites. The AD patients are characterized by elevated levels of copper and zinc in the hippocampal region. These metals have been found to accelerate the accumulation of Aβ and promote the ROS release to the cells. Cu^2+^ acts as a bridge between anionic groups on HSPGs and cationic groups of the NFTs. It was also found that free carboxyl groups are required for HSPG binding to NFTs. Heparin can bind to the extracellular superoxide dismutase, a homotetrameric Cu^2+^ and Zn^2+^ containing glycoprotein, and regulate the activity of these metals which reduce accumulation of Aβ peptides (Figure 6).

## 4. Studying ECM Effects in hiPSC-Derived Forebrain Organoids

Combining the knowledge of forebrain organoids and the ECM development in AD pathology, our studies used heparin (competes for Aβ affinity with HSPG), heparinase III (digests HSPGs), chondroitinase (digests CSPGs), and hyaluronic acid (HA) to treat the cortical and hippocampal forebrain spheroids/organoids exposed to Aβ42 oligomers [166]. Briefly, cortical spheroids were generated using dual SMAD inhibition followed with FGF-2 and cyclopamine (an Shh inhibitor). In the case of hippocampal differentiation, the hiPSCs were treated with dual SMAD inhibitors, cyclopamine, and IWP4 (a Wnt inhibitor), followed by treatment with CHIR99021 (a Wnt activator) and BDNF. Exogenous addition of Aβ42 oligomers was found to reduce the expression of β-tubulin III, suggesting the associated neuronal cell death. The treatments of ECM-related molecules with Aβ42 oligomers promoted β-tubulin III expression, indicating the protective effect of these ECM related molecules. Further analysis on the forebrain and hippocampal markers revealed that ECM enzymes were capable of rescuing TBR1+ cells in cortical forebrain spheroids and PROX1+ neurons in the hippocampal spheroids in the presence of Aβ42 oligomers. These results suggest the necessity of further evaluation of various brain ECMs on forebrain spheroid patterning and cell survival and the related neurological disorders to derive better therapeutic interventions. In addition, heparin-conjugated HA hydrogels were synthesized and characterized [107]. A comparative study with heparin-HA hydrogels with varying modulus reveals that the lower modulus (300–400 Pa) hydrogel supports the forebrain fate whereas the higher modulus (1000–1300 Pa) hydrogel supports hindbrain fate of hiPSCs. This patterning effects may be regulated through modulation of the Hippo/YAP signaling.

## 5. Conclusions and Perspectives

Recently, the findings of adult neurogenesis in humans have sharply declined, which raises questions about animal models to study human central nervous system [167]. The emerging hPSC technology and organoid methods represent promising opportunities to investigate the development of human brain and neurological diseases. These stem cell-based organoid systems have recapitulated some biological events, such as temporal and spatial organization of brain tissues, neuronal-glial cell interactions, and cell-matrix interactions. However, these models need to show promise to reveal new features of normal and pathological phenomena of neurogenesis. One aspect of the interests is that the patient-derived hPSCs with mutations, such as APP or PS1/PS2, could be used to study the roles of specific genes in FAD development and progression. Another aspect is that organoids provide a more advanced in vitro tool to investigate complex niche interactions, such as how HSPGs are involved in Aβ pathogenesis. While the organoid models have great advantages, they are also challenged by the low homogeneity and reproducibility. The brain organoids may be different from each other not only in size but also structure, which makes it difficult to control the quality and predict the readouts. Future improvements, such as novel biomaterials combined with new culture system design (e.g., microfluidics), may be utilized to better control neural patterning in a more accurate and predictable way.

## Figures and Tables

**Figure 1 cells-08-00242-f001:**
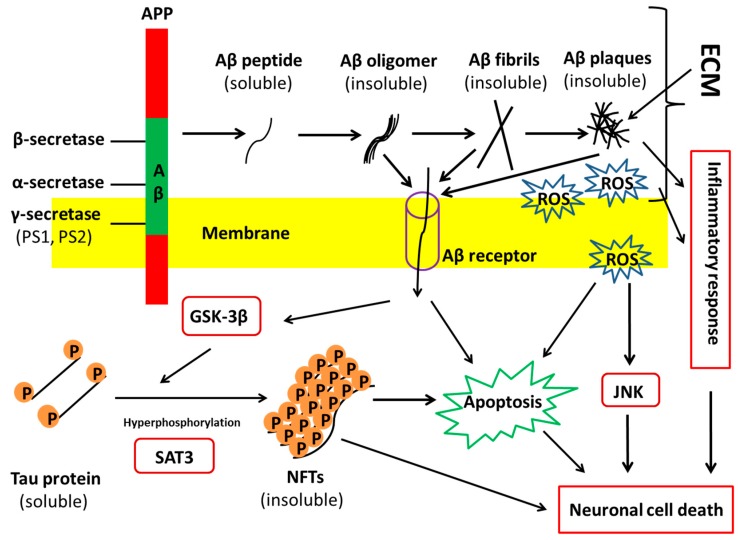
A possible pathology of Alzheimer’s disease (AD). It is postulated that AD may be caused by the deposition of Aβ and tau hyperphosphorylation-derived neurofibrillary tangles (NFTs), both of which could activate the caspase-associated apoptosis. In AD brain the monomeric Aβ peptides aggregate to form toxic Aβ oligomers and further generate the insoluble fibrils, which ultimately form the plaques. The toxic Aβ species could trigger an inflammatory response and increase the level of ROS, which may cause neuron death. On the other hand, toxic Aβ species may be transferred into cells and trigger the apoptosis of neurons. NFTs could either cause neuronal cell death or trigger apoptosis of neurons. Aβ may interplay with tau-derived NFTs formation, while critical questions about Aβ-induced tau pathology in AD are still unanswered. ECM: extracellular matrix; APP: amyloid precursor protein; ROS: reactive oxygen species; SAT3: signal transducer and activator of transcription 3; JNK: c-Jun N-terminal kinase; GSK-3β: glycogen synthase kinase-3β.

**Figure 2 cells-08-00242-f002:**
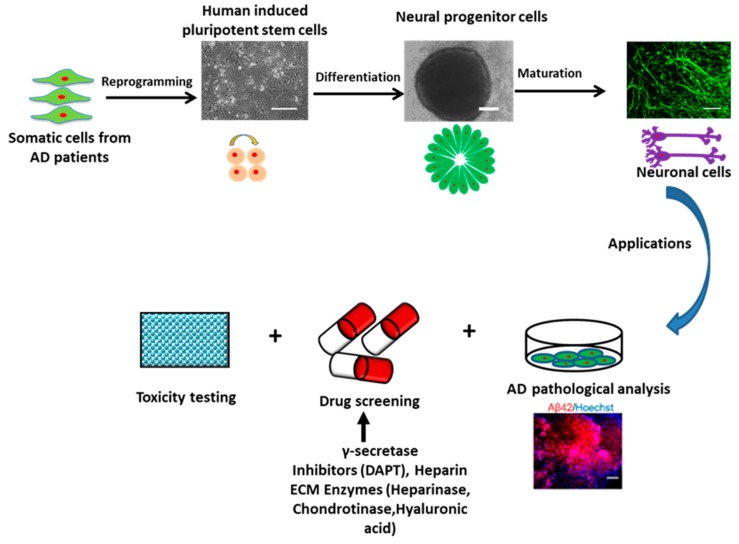
Schematic illustration of the process flow of using hiPSCs for AD studies. The neuronal cells derived from iPSCs cells of AD patients can be used for AD pathological analysis and various therapeutic interventions. AD: Alzheimer’s disease.

**Figure 3 cells-08-00242-f003:**
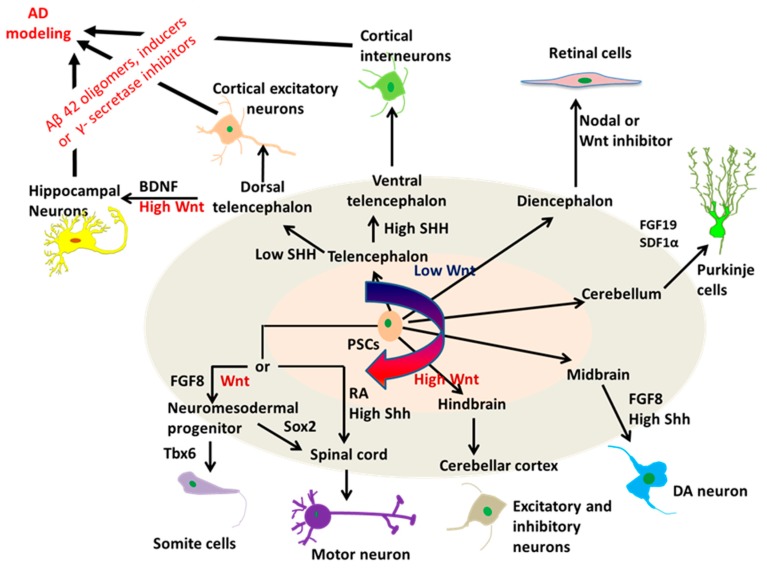
Regional specification in neural differentiation of pluripotent stem cells (PSCs) mimicking in vivo regional patterning, and the use of the derived forebrain organoids for AD modeling. The regional patterning of brain organoids was achieved by cell signaling modulators and the corresponding organoids generated were used for disease modeling. DA: dopaminergic; FGF: fibroblast growth factor; RA: retinoic acid; SHH: Sonic Hedgehog; BDNF: Brain-derived neurotropic growth factor; SDF: Stromal cell-derived factor. Adapted and revised from [81].

**Figure 4 cells-08-00242-f004:**
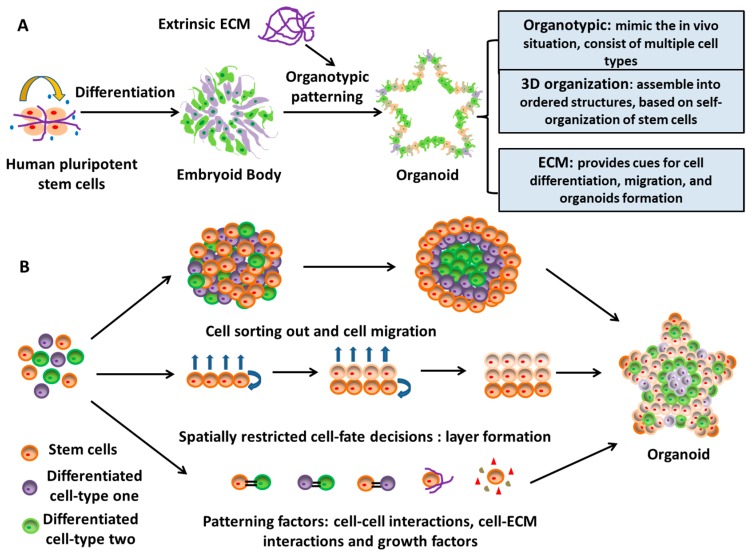
Organoid formation from human pluripotent stem cells. (**A**) Organoids can be generated from human pluripotent stem cells through the embryonic body (EB)-based procedure with the help of exogenous factors, such as extracellular matrix (ECM). Organoids form 3D ordered structures to mimic the in vivo situation of tissues or cells. (**B**) Organoids form based on the self-organization and self-assembly of stem cells.

**Figure 5 cells-08-00242-f005:**
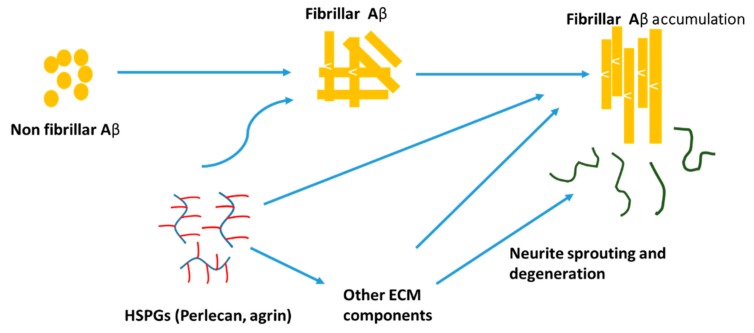
A hypothetical model of heparan sulfate proteoglycan (HSPG) (e.g., agrin/perlecan) involvement in Aβ pathogenesis. ECM: extracellular matrix.

**Figure 6 cells-08-00242-f006:**
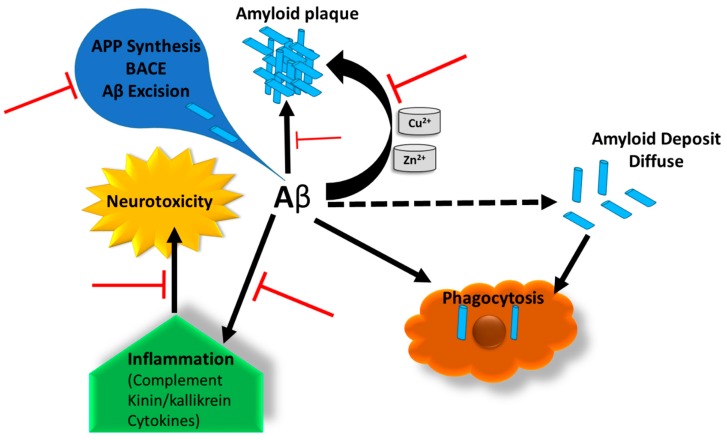
Possible protective actions of heparin in Alzheimer’s disease (AD) pathology. Heparin-based protection can be realized by: (1) reduction of Aβ generation by an action on amyloid precursor protein (APP); (2) prevention of Aβ aggregation/deposition in senile plaques; (3) reduction of the inflammatory response (complement, kinin system); (4) binding to homotetrameric Cu^2+^ and Zn^2+^ containing glycoprotein, and regulating the activity of these metals which reduce accumulation of Aβ peptides. (Red line indicates the actions of heparin on Aβ).

**Table 1 cells-08-00242-t001:** Selected studies of hPSC-based models for Alzheimer’s disease.

Cell Line	Neural Types	AD Phenotypes	Ref.
**Chemicals Induced AD-Phenotypes Using hPSCs**			
Human ESC-derived neurons treated with Aβ42 oligomers	3D neurospheres and 2D, basal forebrain cholinergic neurons expressing ChAT and β-tubulin III	Aβ oligomers suppressed the number of functional neurons	Wicklund et al., 2010 [31]
HiPSC-derived neurons treated with β-secretase (BSI) and γ-secretase inhibitor (GSI) and NSAID	2D, forebrain neurons expressing FOXG1 and TBR1 (62%), CTIP2 (12%), Cux1 (83%) SATB (46%) at day 52	Differentiated neuronal cells expressed Aβ40 and Aβ42. BSI, GSI, and NSAID partially or fully blocked Aβ production in the hiPSCs-derived neuronal cells	Yahata et al., 2011 [32]
Human ESC and hiPSC-derived neurons treated with Aβ42 oligomers	2D, cortical glutamatergic neurons	Aβ oligomers yielded cell culture age-dependent binding of Aβ and cell death in the glutamatergic populations	Vazin et al., 2014 [25]
HiPSC-derived neurons treated with Aβ1-42 oligomers	3D neurospheres, cortical glutamatergic neurons, and motor neurons	Aβ oligomers caused less cell viability, more caspase expression and higher ROS levels on cortical excitatory neurons population. GSK-3β inhibitor may attenuate Aβ-induced cytotoxicity	Yan et al., 2016 [26]
HiPSC-derived neurons induced by Aβ42 inducer (Aftin5)	3D cortical organoids, neurons expressing NeuN, NCAM, MAP2, and CTIP2	Increased secretion of Aβ42 and the Aβ42/40 ratio	Pavoni et al., 2018 [24]
**Overexpression of AD-related gene in hPSCs using genetic modification**			
PSEN1 L166P mutant hPSC-derived neurons treated with γ-secretase inhibitor (DAPT) and NSAID	2D, hPSC-derived neural stem cells (NSCs) expressing Nestin and β-tubulin III	DAPT reduced secretion of both Aβ42 and Aβ40. NSAID reduced secretion of Aβ42. PSEN1 1L166P mutation resulted in elevated Aβ42/40 ratio.	Koch et al., 2012 [27]
PSEN1 (∆E9) mutant hiPSCs	2D, hPSC-derived neural progenitor cells (NPCs) expressing Nestin and tau	The PS1 ∆E9 mutation increases the Aβ42/Aβ40 ratio in human neurons by decreasing Aβ40	Woodruff et al., 2013 [28]
Human ESC-derived neurons model tau pathology	2D, neurons expressing Nestin, DACH1, SOX2, β-tubulin III, and tau	P-tau impaired the transport of mitochondria and led to axonal degeneration and cell death	Mertens et al., 2013 [33]
HPSC-derived neurons co-cultured with ApoE secreted glia	2D, human neurons generated by forced expression of neurogenin-2 (Ngn2), expressing MAP2 and NeuN	ApoE secreted by glia stimulates neuronal Aβ40 and Aβ42 production with an ApoE4 > ApoE3 > ApoE2 potency rank order	Huang et al., 2017 [29]
Human NPCs and hiPSC-derived cells overexpressed APP (K670N/M671L and V717I) mutations	3D microfluidic platform, tri-culture of neurons, astrocytes, and microglia	Increased Aβ aggregation and p-tau formation, induced microglia recruitment and axonal cleavage. Increased chemokines and cytokines.	Park et al., 2018 [30]
**AD patient-derived iPSCs**			
FAD-hiPSCs with PSEN1/2 mutations	2D, neurons expressing β-tubulin III (about 80%) and MAP2	Change in APP processing; increased Aβ42 secretion; responding to γ-secretase inhibitors and modulators.	Yagi et al., 2011 [34]
FAD-hiPSCs from a patient with Down’s syndrome (Trisomy 21 defect)	2D, cortical glutamatergic neurons expressing TBR1, CTIP2, SATB and β-tubulin III	Increased Aβ peptide production, Intracellular and extracellular Aβ42 aggregates. Decreased Aβ40/Aβ42 with γ-secretase inhibitors. Tau hyper-phosphorylation in cell bodies and dendrites. Neuronal cell death.	Shi et al., 2012 [35]
FAD-hiPSCs with APP gene duplications and SAD-hiPSCs	2D, FACS-purified neurons expressing β-tubulin III (>90%) and MAP2	Neurons from AD patients had higher levels of Aβ40, p-tau, and active glycogen synthase kinase-3β (aGSK-3β). β-secretase inhibitors, not γ-secretase inhibitors, reduced p-tau and aGSK-3b.	Israel et al., 2012 [36]
FAD-hiPSCs with APP mutations and SAD-hiPSCs	2D, cortical neurons expressing β-tubulin III, MAP2, TBR1 and SATB2, and astrocytes expressing GFAP	Intracellular Aβ oligomer formation; reduced extracellular Aβ peptides.	Kondo et al., 2013 [37]
FAD-hiPSCs with APP or PSEN1 mutations	2D, neural stem cells (NSCs) expressing Nestin SOX2, ZO1, β-tubulin III, and MAP2	Increased the Aβ42/Aβ40 ratio compared to healthy control. With high concentrations of γ-secretase inhibitors (NSAID-based GSMs drugs), Aβ42/Aβ40 ratio was decreased.	Mertens et al., 2013 [38]
FAD-hiPSCs with PSEN1 mutations	2D, NPCs expressing β-tubulin III	Increased the Aβ42/Aβ40 ratio.	Sproul et al., 2014 [39]
FAD-hiPSCs with PSEN1 (A246E) mutations	3D EB-based, neurons expressing Nestin, PAX6, FOXG1, TBR1, STAB2, β-tubulin III, and MAP2	Increased the Aβ42/Aβ40 ratio, increased expression of FOXG1, mGluR1, and SYT1.	Mahairaki et al., 2014 [40]
FAD-hiPSCs with PSEN1 and AG mutations and SAD-hiPSCs with APOE3/E4 mutations	Basal forebrain cholinergic neurons expressing MAP2, ChAT, and VaChT	Elevated Aβ42. With γ-secretase inhibitors, Aβ40 was increased and calcium transient was increased.	Duan et al., 2014 [41]
FAD-hiPSCs with APP mutations	3D EB-based, forebrain neurons expressing MAP2, tau, β-tubulin III, Cux1, TBR1, vGlut1	Increased Aβ42: Aβ40; Decreased APPsα: APPsβ, γ-secretase inhibitor blocked APPs, β cleavage. Increased total tau and p-tau (Ser262) d100. Aβ antibodies blocked, increased total tau.	Muratore et al., 2014 [42]
FAD-hiPSCs with PSEN1 (A246E, H163R or M146L) mutations	2D, neurons expressing Nestin, PAX6 and SOX1	Increased the Aβ42/Aβ40 ratio compared to healthy control. Reduced Aβ42 and Aβ38 by γ-secretase inhibitor-GSM4.	Liu et al., 2014 [43]
FAD-hiPSCs with PSEN1 mutations	3D EB based, neurons expressing β-tubulin III	Increased the Aβ42 secretion level. Elevated acid sphingomyelinase (ASM) levels. ASM levels restored by ASM siRNA treatment.	Lee et al., 2014 [44]
SAD-hiPSCs with SOR1 variants	2D, FACS-purified neurons expressing Nestin and MAP2	Altered induction of SORL1 expression; altered Aβ peptide production.	Young et al., 2015 [45]
FAD-hiPSCs with PSEN1 or APP mutations	2D, cortical excitatory neurons expressing tau	Increased the Aβ42 secretion level. Increased intracellular tau and phosphorylated tau levels.	Moore et al., 2015 [46]
SAD-hiPSCs with APP mutations	2D, neurons expressing Nestin, PAX6 and β-tubulin III	Increased phosphor-tau (p-tau) and active glycogen synthase kinase-3β (aGSK-3β).Reduced p-tau by γ-secretase inhibitor.	Hossini et al., 2015 [47]
FAD-hiPSCs with PSEN1 (A246E) mutations and SAD-hiPSC mutations	2D, neurons expressing Nestin, SOX2, MAP2, and β-tubulin III	Increased Aβ42 for FAD-hiPSCs-derived neurons.	Armijo et al., 2016 [48]
FAD-hiPSCs with PSEN1 (P117R)/APOE3/3 mutations and SAD-hiPSCs with APOE3/E4 mutations	3D neurospheres, neural cells expressing GFAP, and MAP2	Reduced neurites length and neuronal viability. Elevated levels of nitrite and apoptosis. Hyper-excitable Ca^+^ signaling phenotype. Protected neurites and cell viability by treatment of apigenin.	Balez et al., 2016 [49]
FAD-hiPSCs with APP (V717I) mutations	3D EB based, forebrain neurons expressing GABA, PVB, and MAP2	Elevated levels of Aβ and sAPPα.	Liao et al., 2016 [50]
SAD-hiPSCs	3D neuro-spheroid, cortical neurons expressing PAX6, MAP2, NeuN and β-tubulin III	3D spheroids recapitulated both amyloid β and tau pathology. Reduced Aβ42 and Aβ40 production both in 2D and 3D neurons with BACE1 or γ-secretase inhibitors.	Lee et al., 2016 [51]
FAD-human iPSCs with APP or PS1 mutations	3D brain organoids, neuronal cells expressing SOX2, and MAP2	3D organoids recapitulated amyloid β, tau pathology, and endosome abnormalities. Reduced amyloid and tau pathology with β-and γ-secretase inhibitors.	Raja et al., 2016 [52]
FAD-hiPSCs with PSEN1 (M146L) mutations and SAD-hiPSCs with APOE4 mutations	2D differentiation; cortical neurons and astrocytes	Reduced morphological heterogeneity in astrocytes.	Jones et al., 2017 [53]
FAD-hiPSCs with APP (V717I) mutations	3D EB-based differentiation, caudal neurons expressing HOXB4 and rostral neurons expressing TBR1	Reduced the Aβ42/Aβ40 ratio but increased the Aβ38/Aβ42 ratio for caudal neurons. Higher levels of total and phosphor-tau for rostral neuronal fate.	Muratore et al., 2017 [54]
FAD-hiPSCs with PSEN1 (M146L, G384A, H163R or A246E), APP (V717I) mutations and SAD-hiPSCs with APOE4 mutations	2D, human cortical neurons (iN cells) generated by force expression of neurogenin-2 (Ngn2), iN cells expressing SATB2, MAP2, vGlut1, and TBR2	iPSC-based screening of pharmaceutical compounds for Aβ phenotypes; anti-Aβ cocktail decreased toxic Aβ levels in neurons derived from patients’ cells. A combination of existing drugs synergistically improved Aβ phenotypes of AD.	Kondo et al., 2017 [55]
FAD-hiPSCs with PSEN1 mutations and SAD-hiPSCs with unknown mutations	2D, cholinergic neurons (VAChT), dopaminergic neurons (TH), GABAergic neurons (GAD2/GAD1), and glutamatergic neurons (vGlut1/2)	Increased levels of extracellular Aβ40 and Aβ42 for FAD and SAD samples. Increased the Aβ42/Aβ40 ratio for FAD sample. Increased levels of p-tau and GSK3β.	Ochalek et al., 2017 [56]
FAD-hiPSCs with PSEN1 (∆E9) mutations	3D EB-based differentiation, astrocytes expressing GFAP and S100β	AD astrocytes increased Aβ42 production, altered cytokine release, dysregulated Ca^2+^ homeostasis, increased oxidative stress and reduced lactate secretion.	Oksanen et al., 2017 [57]
FAD-hiPSC with PSEN1 and APP duplication or hiPSCs from Down’s syndrome (Trisomy 21)	2D, cortical neurons expressing TBR1, and MAP2	Synaptic dysfunction (long-term potentiation) caused by PSEN1 and APP duplication secretomes was mediated by Aβ peptides, whereas trisomy 21 neuronal secretomes induced dysfunction through extracellular tau.	Hu et al., 2018 [58]
FAD-hiPSCs with PSEN1 (M146V) mutation	3D cortical organoids, neurons expressing TBR1, SATB2, BRN2, and MAP2	3D organoids recapitulated Aβ, tau pathology, and neuronal cell death. Reduced amyloid β with DAPT, heparin and heparinase.	Yan et al., 2018 [59]
FAD-hiPSC with PSEN1 (A246E) or hiPSCs from Down’s syndrome (Trisomy 21)	3D cortical organoids, neurons expressing NeuN, SATB2, TBR1, and MAP2	Accumulation of Aβ and tau aggregates and induced cellular apoptosis AD organoids.	Gonzalez et al., 2018 [60]
SAD-hiPSCs from APOE4/E3 mutations	3D organoids, neurons, astrocytes, and microglia-like cells	APOE4 organoids displayed increased Aβ aggregation and hyperphosphorylation of tau.	Lin et al., 2018 [61]
SAD-hiPSCs from unknown mutations	3D neuro-spheroid, neurons	AD organoids neuronal dysfunction was similar to AD brain tissue by mass spectrometry-based proteomics analysis.	Chen et al., 2018 [62]
SAD-hiPSCs from APOE4/E3 mutations	2D, neuronal cells expressing MAP2	Showed aberrant mitochondrial function. Increased levels of ROS and DNA damage. Increased levels of oxidative phosphorylation chain complexes.	Birnbaum et al., 2018 [63]
FAD-hiPSCs and SAD-hiPSCs	2D, FACS-purified neurons	Reduced tau phosphorylation by retromer stabilization.	Young et al., 2018 [64]
HiPSCs from a Down’s syndrome patient by controlling APP gene copy number	2D, cortical neurons	Higher APP gene dosage increased Aβ production, altered the Aβ42/Aβ40 ratio and caused deposition of the pyroglutamate (E3)-containing amyloid aggregates.	Ovchinnikov et al., 2018 [65]
SAD-hiPSCs from APOE4/4 or APOE3/3 mutations	2D, cortical neurons and GABAergic neurons	APOE4 increased Aβ production in human neurons, APOE4-expressing neurons had higher levels of tau phosphorylation.	Wang et al., 2018 [66]
FAD-hiPSCs with APP duplication mutants	2D, FACS-purified neurons	Neuronal cholesteryl esters (CE) regulated the proteasome-dependent degradation of p-tau, CE-mediated Aβ secretion by a cholesterol-binding down in APP, A CYP46A1-CE-tau axis was identified as an early pathway.	van der Kant et al., 2019 [67]

Note: PSCs, pluripotent stem cells; ESCs, embryonic stem cells; iPSCs, induced pluripotent stem cells; AD, Alzheimer’s disease; Aβ, β-amyloid peptide; 2D, two-dimensional; 3D, three-dimensional; BSI/BACE1, β-secretase inhibitor; GSI, γ-secretase inhibitor; NSAID, nonsteroidal anti-inflammatory drug; ROS, reactive oxygen species; GSK-3β, glycogen synthase kinase 3 beta; NSCs, neural stem cells, NPCs, neural progenitor cells; PSEN1/2, presenlin1/2; APP, amyloid precursor protein; FAD, familial AD; SAD, sporadic AD; EB, embryoid body; FACS, fluorescence-activated cell sorting; ChAT: Choline Acetyltransferase. Other Useful studies: Hu et al., 2015 Cell Stem Cell [68], Human chemical-induced neuronal cells (hciNs) from FAD patient fibroblasts with APP (V717I) or PSEN1 (I167del or A434T or S169del) mutations, increased extracellular Aβ42 level and the Aβ42/Aβ40 ratio. Espuny-Camacho et al., 2017 Neuron [69], Chimeric model of AD generated using hPSCs-derived neurons (hPSC-neurons grafted into AD mice), major degeneration and loss of human neurons in chimeric AD mice, absence of tangle pathology in degenerating human neurons in vivo. Wang et al., 2017 Stem Cell Reports [70], Neurogenin 2 (NGN2)-induced glumatergic neurons (iN cells) from hiPSCs, iN cells are used to identify tau-lowering compounds in LOPAC (Library of Pharmacologically Active Compounds), and identified adrenergic receptors agonists as a class of compounds that reduce endogenous human tau.

**Table 2 cells-08-00242-t002:** Effects of matrix modulus on pluripotent stem cell fate decisions.

Cell Source	Range of Modulus and Substrates	Effect on Morphology, Proliferation, and Differentiation	Reference
Neural progenitor cells	0.1 kPa–10 kPa; PA gels based vmIPNs	Soft gel (100–500 Pa) favored neurons, harder gel (1–10 kPa) promoted glial cells.	Saha et al., 2009 [111]
Neural progenitor cells	1–20 kPa; MAC substrates	<1 kPa favored neuronal differentiation; <3.5 kPa supported astrocyte, >7kPa favored oligodendrocyte.	Leipzig et al., 2009 [112]
Mouse ESCs	41–2700 kPa; collagen coated PDMS surface	Increasing substrate stiffness from 41–2700 kPa promoted cell spreading, proliferation, mesendodermal and osteogenic differentiation.	Evans et al., 2009 [122]
Rat neural stem cells	180–20,000 Pa; 3D alginate hydrogel scaffolds	The rate of proliferation of neural stem cells decreased with an increase in the modulus of the hydrogels. Lower stiffness enhanced neural differentiation.	Banerjee et al., 2009 [123]
Mouse ESCs	0.6 kPa; PA gel substrates	Soft substrate supported self-renewal	Chowdhury et al., 2010 [124]
Human ESCs and iPSCs	0.7–10 kPa; GAG-binding hydrogel	The stiff (10 kPa) hydrogel maintained cell proliferation and pluripotency.	Musah et al., 2012 [125]
Human ESCs	0.05–7 MPa, 3D PLLA, PLGA, PCL or PEGDA scaffold coated with matrigel	50 to 100 kPa supported ectoderm differentiation; 100 to 1000 kPa supported endoderm differentiation; 1.5 to 6 MPa supported mesoderm differentiation.	Zoldan et al., 2011 [126]
Human ESCs and iPSCs	0.1–75 kPa; matrigel-coated PA gels	Soft matrix (0.1 kPa) promoted early neural differentiation.	Keung et al., 2012 [119]
Human ESCs	1 kPa, 10 kPa, 3 GPa; PDMS substrates	Rigid matrix promoted cardiac differentiation.	Arshi et al., 2013 [127]
Mouse ESCs	0–1.5 kPa, 3D collagen-I, Matrigel, or HA hydrogel	<0.3 kPa less neurite outgrowth and supported glial cell; 0.5 to 1 kPa more neurite outgrowth and supported neurons.	Kothapalli et al., 2013 [113]
Human ESCs	0.078–1.167 MPa; PDMS substrates	Increased stiffness upregulated mesodermal differentiation.	Eroshenko et al., 2013 [128]
Human ESCs	1.3 kPa, 2.1 kPa, 3.5 kPa; HA hydrogel	Stiffness of 1.2 kPa was the best to support pancreatic β-cell differentiation.	Narayanan et al., 2014 [129]
Human ESCs	4–80 kPa; PA hydrogels	Stiffness of 50 kPa was the best for cardiomyocyte differentiation. Stiffness impacted the initial differentiation of hESCs to mesendoderm, while it did not impact differentiation of cardiac progenitor cells to cardiomyocytes.	Hazeltine et al., 2014 [130]
Human iPSCs	19–193 kPa; 3D PCL, PET, PEKK or PCU electrospun fibers	The substrate stiffness was inversely related to the sphericity of hiPSC colonies.	Maldonado et al., 2015 [131]
HPSCs	6 kPa, 10 kPa, 35 kPa; Matrigel micropatterns	High stiffness (35 kPa) induced myofibril defects of hPSC-derived cardiomyocytes and decreased mechanical output.	Ribeiro et al., 2015 [132]
hPSC-derived hepatocytes (hPSC-Heps)	20, 45, 140 kPa; collagen-coated PA hydrogels substrates	On softer substrates, the hPSC-Heps formed compact colonies while on stiffer substrates they formed a diffuse monolayer. Albumin production correlated inversely with stiffness.	Mittal et al., 2016 [133]
Rat cortical neurons (RCN)	5 kPa (soft), PA gels; 500 kPa (stiff), PDMS substrates;	Soft substrates enhanced cortical neurons migration. Stiff substrates increased synaptic activity.	Lantoine et al., 2016 [114]
Mouse ESCs and iPSCs	300–1200 Pa; 3D PEG hydrogels	Stiffness and other biophysical effectors promoted somatic-cell reprogramming and iPSC generation; lower modulus (300–600 Pa) showed higher reprogramming efficiency.	Caiazzo et al., 2016 [134]
Human ESCs	400 Pa, 60 kPa; PA hydrogels	On stiff substrates, β-catenin degradation inhibits mesodermal differentiation of human ESCs.	Przybyla et al., 2016 [135]
Human ESCs	1–100 kPa; barium alginate capsules	Stiffness of 4–7 kPa supported cell proliferation and higher stiffness suppressed cell growth. Increased stiffness promoted endoderm differentiation, while suppressed pancreatic induction. About 3.9 kPa was the best for pancreatic differentiation.	Richardson et al., 2016 [136]
Mouse intestinal stem cells (ISC)	300 Pa, 700 Pa, 1.3 kPa, 1.7 kPa; PEG hydrogels	Higher stiffness enhanced ISC expansion. Lower stiffness supported ISC differentiation and organoid formation.	Gjorevski et al., 2016 [137]
Mouse neural progenitor cells (NPC)	0.5–50 kPa; 3D elastin-like protein hydrogels	In stiffness from 0.5 to 50 kPa, NPC stemness maintenance did not correlate with initial hydrogel stiffness.	Madl et al., 2017 [115]
Mouse ESCs and hiPSCs	10–100 kPa; 3D PU scaffolds	Scaffolds with proper stiffness, Poisson’s ratio and pore structure enhanced neural differentiation of PSCs.	Yan et al., 2017 [108]
Human iPSCs	3–168 kPa; PDMS substrates	Elasticity of substrates significantly affected cell colony formation. Intermediate substrate elasticity of about 9 kPa is preferable to reach an EB-like aggregation and optimal for cardiac differentiation.	Wang et al., 2018 [138]
Mouse ESCs	3.4 kPa, 64 kPa, 144 kPa; PEGDA or PEG hydrogel substrates	Soft hydrogel (3.4 kPa) showed strong cell attachment and a growth pattern similar to 2D surface. Stiff hydrogel (144 kPa) supported a 3D aggregation.	Dorsey et al., 2018 [139]
Mouse iPSCs	0–2.4 MPa; PDMS substrates	Stiffer substrate supported pluripotency of iPSCs. Softer substrate promoted cardiac differentiation.	Fu et al., 2018 [140]
Neural crest stem cells (NCSCs) from hiPSCs	1kPa, 15 kPa, 1 GPa; PA gel substrates	>50 kPa promoted smooth muscle cells from NCSCs, <15 kPa promoted glial cells from NCSCs.	Zhu et al., 2018 [116]
Mouse hippocampal neurons	2.13 kPa, 22.1 kPa; PDMS substrates	Stiff substrate enhanced voltage-gated Ca^2+^ channel currents in neurons.	Wen et al., 2018 [117]
Neural crest stem cells (NCSCs) derived from hESCs	3.3 kPa, 1.7 MPa, 1 GPa; PDMS substrates	Soft substrate increased differentiation of ectodermal mesenchymal stem cells (MSCs) from NCSCs via CD44 mediated PDGFR signaling.	Srinivasan et al., 2018 [118]
iPSCs and neonatal rat cardiomyocytes	9, 20, 180 kPa; PA gel substrates	Cardiac differentiation preferred rigid substrates, and beating behavior preferred soft substrate.	Hirata et al., 2018 [141]
Human iPSCs	About 24 Pa, fibrin-based gel substrates (human platelet lysate gel); >1 GPa, tissue culture plastics	Soft substrates did not impact on differentiation of iPSCs into MSCs.	Goetzke et al., 2018 [142]
Human ESCs	118 ± 51 Pa, 800 ± 180 Pa, 5600 ± 1100 Pa, and 8900 ± 1500 Pa; decellularized fibroblast-derived matrices crosslinked by genipin	Soft matrix supported cell migration and induced EMT of hPSC. Stiff matrix supported cell pluripotency and suppressed EMT of hPSCs.	Kim et al., 2018 [143]

Note: PA, polyacrylamide; vmIPNs, variable moduli interpenetrating polymer networks; MAC, methacrylamide chitosan; PDMS, polydimethylsiloxane; HA, hyaluronic acid; PLLA, poly-L-lactide acid; PLGA: poly(lactic co-glycolic acid); PCL, polycaprolactone; PEGDA, polyethylene glycol diacrylate; PET, polyrethylene terephthalate; PEKK, poly(etherketoneketone); PCU, polycarbonate-urethane; GAG, glycosaminoglycan; PEG, polyethylene glycol; PU, polyurethane; PDGFR, platelet-derived growth factor receptor beta (PDGFRβ) signaling; EMT, epithelial-mesenchymal-transition.

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
