# Peer review of "The Use of Pluripotent Stem Cell-Derived Organoids to Study Extracellular Matrix Development during Neural Degeneration"

_cells, 2019, doi:10.3390/cells8030242_

Round 1

Reviewer 1 Report

In this MS, Yan Y et al. reviewed the AD pathologies and also comprehensively summarized all the current AD models. They also underlined the important roles of ECM in brain function, AD and the other brain diseases. Overall, the content provide by this MS is interesting and important for the readers in this field. I am very impressive by the details of hPSC-based models for AD and effects of matrix on the fate decision of stem cells.

Major concerns:

1.   The figures are well prepared, but most figure legends do not describe the figures in detail.

2.   The content in 1.1 (line 34-46) seems to be less correlated with AD pathology.

3.   Please clarify ROS, SAT3, NET… in Figure 1.

4.   The representation of Figure 2 is not clear. Do the applications include AD modeling, Drug screening…?

5.   Figure 3 has been reported in previous review. The problem here is that the author did not added any current advances into this figure, meaning that there is no true contribution.

Author Response

Major concerns:

1.     The figures are well prepared, but most figure legends do not describe the figures in detail.

Response: As the reviewer suggested, we revised Figure Legends for Figure 1-3.

2.   The content in 1.1 (line 34-46) seems to be less correlated with AD pathology.

Response: As the reviewer suggested, we removed some of background information in Introduction. First four sections (1.1, 1.2, 1.3, and 1.4) were combined together as one section.

3.   Please clarify ROS, SAT3, NET… in Figure 1.

Response:  As the reviewer suggested, we clarified ROS, SAT3, JNK, GSK in Figure 1 legend.

4.   The representation of Figure 2 is not clear. Do the applications include AD modeling, Drug screening…?

Response:  As the reviewer suggested, we revised Figure 2.

5.   Figure 3 has been reported in previous review. The problem here is that the author did not added any current advances into this figure, meaning that there is no true contribution.

Response:  As the reviewer suggested, we revised Figure 3.

Reviewer 2 Report

The manuscript by Yan et al, explore the possible use of pluripotent stem cell-derived organoids to study extracellular matrix development during neural degeneration for Alzheimer’s disease.

The manuscript is well written and easy to follow, and the pictures properly summarize the literature review. However, the background given about the Alzheimer’s disease is too general. I suggested the authors to skip directly to the point of their work.

Author Response

The manuscript by Yan et al, explore the possible use of pluripotent stem cell-derived organoids to study extracellular matrix development during neural degeneration for Alzheimer’s disease.

The manuscript is well written and easy to follow, and the pictures properly summarize the literature review. However, the background given about the Alzheimer’s disease is too general. I suggested the authors to skip directly to the point of their work.

Response: As the reviewer suggested, we removed some of background information in Introduction. First four sections (1.1, 1.2, 1.3, and 1.4) were combined together as one section.